# Mapping of Metabolic Heterogeneity of Glioma Using MR-Spectroscopy

**DOI:** 10.3390/cancers13102417

**Published:** 2021-05-17

**Authors:** Pamela Franco, Irene Huebschle, Carl Philipp Simon-Gabriel, Karam Dacca, Oliver Schnell, Juergen Beck, Hansjoerg Mast, Horst Urbach, Urs Wuertemberger, Marco Prinz, Jonas A. Hosp, Daniel Delev, Irina Mader, Dieter Henrik Heiland

**Affiliations:** 1Department of Neurosurgery, Medical Center-University of Freiburg, 79106 Freiburg, Germany; irene.huebschle@gmail.com (I.H.); k.dacca@gmail.com (K.D.); oliver.schnell@uniklinik-freiburg.de (O.S.); j.beck@uniklinik-freiburg.de (J.B.); dieter.henrik.heiland@uniklinik-freiburg.de (D.H.H.); 2Faculty of Medicine, University of Freiburg, 79106 Freiburg, Germany; carl.philipp.simon-gabriel@uniklinik-freiburg.de (C.P.S.-G.); horst.urbach@uniklinik-freiburg.de (H.U.); urs.wuertemberger@uniklinik-freiburg.de (U.W.); marco.prinz@uniklinik-freiburg.de (M.P.); jonas.hosp@uniklinik-freiburg.de (J.A.H.); IrMader@schoen-klinik.de (I.M.); 3Department of Radiology, Medical Center-University of Freiburg, 79106 Freiburg, Germany; 4Department of Neuroradiology, Medical Center-University of Freiburg, 79106 Freiburg, Germany; hansjoerg.mast@uniklinik-freiburg.de; 5Institute of Neuropathology, Faculty of Medicine, University of Freiburg, 79106 Freiburg, Germany; 6Signaling Research Centers BIOSS and CIBSS, University of Freiburg, 79106 Freiburg, Germany; 7Center for Basics in NeuroModulation (NeuroModulBasics), Faculty of Medicine, University of Freiburg, 79106 Freiburg, Germany; 8Department of Neurology and Neuroscience, Medical Center-University of Freiburg, 79106 Freiburg, Germany; 9Department of Neurosurgery, RWTH University of Aachen, 52074 Aachen, Germany; ddelev@ukaachen.de; 10Specialist Centre for Radiology, Schoen Clinic, 83569 Vogtareuth, Germany; 11Microenvironment and Immunology Research Laboratory, Medical Center-University of Freiburg, 79106 Freiburg, Germany

**Keywords:** radiomics, magnetic resonance spectroscopy, MRS, MR spectroscopy, chemical shift imaging, 1H-MRS, glioma, neuroradiology, neurosurgery, neurooncology

## Abstract

**Simple Summary:**

Radiomics is a research field that integrates radiological and genetic information, but the application of the techniques that have been developed to this purpose have not been widely established in daily clinical practice. The purpose of our study is the development of a straightforward tool that can easily be used to preoperatively predict and correlate the metabolic signature of different CNS-lesions. Particularly in gliomas, we hope to integrate the molecular profile of these tumors into our prediction model. Our goal is to deliver an open-software tool with the intention of advancing the diagnostic work-up of gliomas to the latest standards.

**Abstract:**

Proton magnetic resonance spectroscopy (^1^H-MRS) delivers information about the non-invasive metabolic landscape of brain pathologies. ^1^H-MRS is used in clinical setting in addition to MRI for diagnostic, prognostic and treatment response assessments, but the use of this radiological tool is not entirely widespread. The importance of developing automated analysis tools for ^1^H-MRS lies in the possibility of a straightforward application and simplified interpretation of metabolic and genetic data that allow for incorporation into the daily practice of a broad audience. Here, we report a prospective clinical imaging trial (DRKS00019855) which aimed to develop a novel MR-spectroscopy-based algorithm for in-depth characterization of brain lesions and prediction of molecular traits. Dimensional reduction of metabolic profiles demonstrated distinct patterns throughout pathologies. We combined a deep autoencoder and multi-layer linear discriminant models for voxel-wise prediction of the molecular profile based on MRS imaging. Molecular subtypes were predicted by an overall accuracy of 91.2% using a classifier score. Our study indicates a first step into combining the metabolic and molecular traits of lesions for advancing the pre-operative diagnostic workup of brain tumors and improve personalized tumor treatment.

## 1. Introduction

With the revised WHO-Classification of Tumors of the Central Nervous System of 2016 it was made clear that the molecular biology of the tumors surpasses in importance the histological features, due to the fact that the first is crucial not only for diagnosis, but also for prognosis in terms of response to adjuvant therapy and survival [1]. Parallel to these developments, the appeal of thoroughly exploring brain tumors from a neuroradiological point of view has led to important advances in the preoperative classification of different brain tumors through radiomics, a concept for feature-based decomposition of MRI data. Numerous algorithms and studies were performed using artificial intelligence and novel machine-learning approaches, but a translation into clinical routine is pending. The need for precise presurgical imaging tools to classify the tumor origin or for detailed information regarding molecular configuration is high. The ultimate goal is a more personalized approach in the overall oncological treatment of patients with brain tumors, with physicians being able to better plan a surgical treatment, being biopsy or surgical resection, and the extent and complexity of it as well as preparing the patient for the adjuvant treatment afterwards. Additionally, such data can improve therapy monitoring and allows for early detection of recurrences. Besides classical MRI imaging analysis, MRS was found to be an accurate tool for gaining insights into the molecular biology of brain tumors. Choi et al. were able to detect (R)-2-HG, the oncometabolite generated by the IDH 1/2 mutation, which could be used for diagnostic proposes or monitoring of therapy [2]. Previous studies including those by our team have shown a successful way in which MRS can predict the molecular profile of gliomas based on the metabolic alterations [2,3,4,5]. In pediatric cerebellar tumors, a multicenter study has validated the importance of adding MRS to the routine MRI diagnostic workup and showed the prognostic value of the biomarkers identified through imaging and also developed an automated tool implemented in the automatic analysis of peak metabolite ratios and the classification of different pathologies [6,7,8]. The importance of developing automated analysis tools lies in the possibility of a straightforward application and simplified interpretation of metabolic and genetic data that allow for incorporation into the daily practice of a broad audience. Here, we present our prospective imaging trial and the correlation between the metabolic and molecular profiles of different brain pathologies across varied ages, which we used for characterization and training of an automated tool.

## 2. Methods

### 2.1. Ethical Approval

The local ethics committee of the University of Freiburg approved the data evaluation, imaging procedures and experimental design (protocol 100020/09 and 472/15_160880). The methods were carried out in accordance with the approved guidelines. The study has been approved by the Ethics Committee of the Medical Centre University Freiburg (protocol 360/16_170908) and has been registered at the German Clinical Trials Register (DRKS) under DRKS00019855.

### 2.2. Patient Enrolment

Our prospective, single center, single arm, diagnostic study enrolled from August 2016 until October 2019 120 patients across all ages who were undergoing routine diagnostic imaging for neurologic symptomatic suggestive of brain lesion and matched following criteria: (1) brain lesion not located near the skull base, (2) no contraindication for undergoing MRI, (3) no contrast-enhancement allergy. All subjects or their respective legal guardians, in the case of one patient younger than 18 years of age, signed a written informed consent for participation in the study. Following the acquisition of imaging, patients underwent either biopsy or surgical resection for their respective lesions after previous discussion of the case by our center’s multidisciplinary tumor board.

### 2.3. Imaging Acquisition and Data

Preoperative anatomical and in-vivo chemical shift imaging were performed at a whole-body system 3T MR Magnetom Prisma scanner (Siemens, Erlangen, Germany) in the Department of Neuroradiology, Medical Centre-University Freiburg. Anatomical imaging included a 3D SPACE Dark Fluid sequence (TR = 5000 ms, TI = 1800 ms, TE = 388 ms, echo train length 251, voxel size 1 mm^3^), a 2D T2 weighted turbo spin-echo sequence (TR = 4500 ms, TE = 100 ms, echo train length 17, voxel size 0.6 × 0.6 × 2 mm^3^) in the geometry of the CSI-sequence, and a 3D T1-weighted sequences (MPRage, TR = 2300 ms, TI = 988 ms, TE = 2.26 ms, voxel size 1 mm^3^) before and after application of 0.1 mM Gadoteridol per kg body weight [ProHance^®^, Bracco, Konstanz, Germany]. Spectroscopic imaging was performed using the manufacturer’s provided 2D chemical shift imaging (CSI) sLASER sequence with a TR = 1500 ms, TE = 40 ms, FOV 160 × 160 mm^2^, phase enconding steps 32 × 32, resulting in a nominal voxel size of 5 × 5 × 20 mm^3^.

### 2.4. Data Processing

Raw spectra were downloaded in the RDA file format (Siemens) and loaded into R using the package “spant” [9]. We further analyzed the data by an internal pipeline including baseline correction and fitting using Totally Automatic Robust Quantitation in NMR (TARQUIN, http://tarquin.sourceforge.net, accessed on 5 March 2021). The full pipeline is available at github.com/heilandd/SPORT/R/Pipeline.R, accessed on 5 March 2021. Data were exported into an RDS file (R-file format) containing the raw baseline corrected spectra as well as the TARQUIN output. A detailed description is given in the Appendix A.

### 2.5. Segmentation

We designed an interactive “shiny” app for “R” (https://shiny.rstudio.com) for a semi-automated segmentation. First, the MRS grid was aligned to the FLAIR-, T2- and contrast-enhanced T1-weighted images. The segmentation assigned regions into normal appearing matter (NAM), lesion (i.e., ischemia, inflammation), tumor, contrast, necrosis, FLAIR hyperintense region or ventricle system. Each spectrum was supervised and excluded if quality criteria based on fitted residuals was not achieved. Segmentation and raw files were exported for further analysis.

### 2.6. Tumor Tissue Sampling

Tumor tissue was obtained by either stereotactic biopsy or surgical resection at the Department of Neurosurgery of the Medical Center- University Freiburg. Tissue samples of the contrast-enhancement region were immediately snap-frozen in liquid nitrogen and processed. Histopathological diagnosis was performed at the Institute of Neuropathology, Medical Center-University Freiburg according to their standards, including IDH immunohistochemistry and genome analysis (1p19q co-deletion (LOH) and exome sequencing of IDH1/2).

### 2.7. Deep Autoencoder for Denoising

For data denoising, we used the autoencoder framework to estimate a gaussian distribution conditioned on the input matrix containing 34 metabolites. For the input layer, we normalized and scaled the metabolites: $norm\left(x\right)=\frac{x-\min\left(x\right)}{\max\left(x\right)-\min\left(x\right)}$. The autoencoder consists of two parts: an encoder and a decoder, which can be defined as transitions:(1) encoder: ϕ:X→F decoder: ψ:F→X
(2)$$\phi ,\psi =arg.min\|X-\left(\phi ,\psi \right)X{\|}^{2}$$

A sigmoid activation function with a minimal dropout (b = 0.1) was recently described as being beneficial for precise data reconstruction and denoising [10] (Appendix A), thus we defined our activation function as follows:(3)$$\sigma \left(x\right)=\frac{1}{1+e^{-x}}$$

The encoder stage of an autoencoder takes the input x ∈ℝd=X  and maps it to z ∈ℝp=F at the layer position φ:(4)x=Aφ=0 ; zφ=σWφ×Aφ−1+bφ
zφ is also referred to as *latent representation*, here presented as z1, z2, …, zφ=n  in which φ describes the number of hidden layers. **W** is the weight matrix and **b** represents the dropout or bias vector. Our network architecture contained 3 hidden layers including a batch normalization with decreasing number of neurons (32, 16, 8), followed by a bottleneck layer of neurons (6) and the decoder part with mirrored architecture. In the decoder, weights and biases are reconstructed through backpropagation (ψ:F→X) and *z* is mapped to x′=A0′ in the shape as x′:(5) Aφ−1′=σ′Wφ′×zφ+bφ′

In this context, $W^{\prime },\;\sigma^{\prime },\;b^{\prime }$ from the decoder are unrelated to W,σ ,b from the encoder. We used a loss function to train the network in order to minimize reconstruction errors.
(6)$$\;L\left(x,x\prime \right)=\|x-\sigma \prime \left(W^{\prime }\left(\sigma \left(Wx+b\right)\right)+b^{\prime }\right){\|}^{2}$$

The following hyperparameters were used: Optimizer: ‘adam’, training/test split 2/3, epochs: 500.

### 2.8. Hyperparameter Search

In order to determine the optimal number of bottleneck and dropout values, we trained the network with an increasing number of bottleneck layers (*n* = 4:10) and determined the variance of the two first eigenvectors as a benchmark to determine the maximum effectiveness of the denoising algorithm.

### 2.9. Prediction Model

For prediction of the voxel origin (normal appearing matter vs. lesion) and further prediction of the tumor subclass, we used a linear discriminant analysis (LDA) as our denoised output (deep autoencoder) is normally distributed. We defined our metabolic training data as a set of observations $\underset{x}{\to }$ for each voxel with the known class μ. LDA addresses the problem by postulating that the conditional probability density functions are normally distributed when considering both mean and covariance parameters. Our input was defined as:(7)$$\;x^{\prime }=\sigma^{\prime }\left(W^{\varphi }\prime \times z^{\varphi }+b^{\varphi }\prime \right)\;\sim \;N\left(µ,\sigma^{2}\right)$$
$x^{\prime }$ is the reconstructed vector x using the autoencoder. In addition, we made a simplifying homoscedasticity assumption that the covariances have full rank due to the deep autoencoder outputs. Here, we applied a multiclass LDA which is defined by:(8)$$S=\frac{\underset{w}{\to }T\times \sum_{b}\underset{w}{\to }}{\underset{w}{\to }T\times \sum \underset{w}{\to }}$$
(9)$$\sum_{b}=\frac{1}{C}\sum_{i=1}^{C}\left(\mu_{i}-\mu \right){\left(\mu_{i}-\mu \right)}^{T}$$
in which $\underset{w}{\to }$ is the eigenvector and C represents the classes (C = C_NAM,_ C_T,_ C_L_), and for the tumor classification: C = C_wt,_ C_IDH,_ C_LOH_. The final classification score (SPORT classifier) is the mean of the estimated class predictors (LDA) resulting in a score for each determined class:(10)$$\;s_{c\left(n\right)}=mean(\sum_{i}^{C}S\left(c\right))$$

### 2.10. Dimensional Reduction and Clustering

We decomposed eigenvalue frequencies of the first 30 principal components (PCs) and determined the number of non-trivial components by comparison to a matrix containing randomized intensity values. As non-trivial eigenvectors, we defined all PCs demonstrating a larger variance compared to the random matrix. The obtained non-trivial components were used for shared nearest neighbor (SNN) clustering followed by dimensional reduction using the Uniform Manifold Approximation and Projection (UMAP [11]). All steps were implemented in our software package in R. The functions and downstream pipeline can be accessed in GitHub (San Francisco, CA, USA; https://github.com/heilandd/SPORT-STUDY, accessed on 2 September 2020) in the “package.R” file. A short tutorial is given in the Appendix A.

### 2.11. Spatial Data Analysis

For the analysis of the spatial distribution, we optimized our SPATA [12] toolbox (spatial transcriptomic analysis) by using an MRS input function to create a “spata” S4 object. SPATA allows for determination of marker metabolites in space, trajectory analysis and metabolic architecture. A full example and the input code are available at GitHub: https://themilolab.github.io/SPATA/index.html (accessed on 2 September 2020).

### 2.12. Mean Spectra

We estimated all presented sum spectra by the mean and variance of the baseline corrected raw spectra. Spectra were plotted by the ppm (*x*-axis) and the intensity (*y*-axis). The plotted lines indicated the mean signal intensity. The spectra were heat-colored representing the regions of most variance using the inferno color scheme: yellow indicates strong and black indicates low variance.

### 2.13. Data Availability

We can provide full anonymized RDA files upon reasonable request. Data are available as an RDS file containing segmentation, some clinical information (anonymization and confidentially are secure according to European laws) and raw and fitted spectra.

## 3. Results

### 3.1. Patient Demographics

The enrolment of patients in our prospective imaging trial started in August of 2016 and 120 patients were enrolled until October 2019. From the 120 patients recruited, 23 were found to have deviations from the study protocol, including erroneous MRS parameters (*n* = 8), bad shimming (*n* = 2) and wrong voxel size (*n* = 13). Additionally, we excluded six patients whose diagnosis was either clinically and/or histologically uncertain or who were lost to follow-up after imaging. We included 91 datasets from 90 patients in our analysis (one patient was imaged at diagnosis and first recurrence). All patients received a chemical-shift imaging of 256 voxel (5 × 5 × 20 mm^3^), Appendix A. The cohort was well balanced regarding age, tumor/lesion localization and gender, Appendix A. The most frequent pathology was glioma, representing 73% (*n* = 65) of all patients. Within gliomas, those with a wildtype IDH-status were more common (*n* = 35, 53.8%, histopathologically defined glioblastomas, WHO grade IV), followed by those with a mutated IDH-status (*n* = 17, 26.2%, histopathologically defined as diffuse and anaplastic astrocytoma, WHO grades II and III respectively) and lastly there were *n* = 13 patients with a mutated IDH-status combined with a codeletion of 1p/19q (LOH, 20%, histopathologically defined as oligodendrogliomas WHO grades II and III), Appendix A. We also included a broad spectrum of different pathologies including dysembrioplastic neuroepithelial tumors (DNET), ependymoma, pineal gland tumors, metastatic lesions from melanoma and nonsmall-cell lung carcinoma (NSCLC), focal cortical dysplasia (FCD) and others, Appendix A.

### 3.2. Unsupervised Analysis of Metabolomic Heterogeneity and Diversity in MRS Data

In order to gain insights into the diversity and spatial distribution of 22,438 proton spectra (obtained from 91 ^1^H-MRS studies), we designed a pipeline for unsupervised analysis using computational approaches predominantly applied to high-dimensional data analysis. After the MRS data acquisition, we fitted the metabolites of each voxel using Totally Automatic Robust Quantitation in NMR (TARQUIN) and segmented each voxel into normal appearing matter (NAM), ventricle, FLAIR-hyperintense region, non-contrast enhancing tumor and contrast enhancing tumor. Voxels that couldn’t be allocated in a straightforward matter or contained overlapping regions, were excluded from the initial data analysis to ensure that the training data set was as accurate as possible. We denoised our data using a deep autoencoder with five bottleneck layers, which revealed an improved data quality, Figure 1a and Appendix A. We used hyperparameter optimization to estimate the dropout in the hidden layers and measured the optimal number of bottleneck layers, Appendix A. After denoising, we performed a principal component analysis (PCA) and selected non-trivial eigenvectors with higher eigenvalues compared to a PCA of randomized metabolic intensities (detailed information in the Section 2), Figure 1b. The first twelve components were used for UMAP, a common method for dimensional reduction. We then performed a shared-nearest-neighbor clustering and identified ten different clusters, Figure 2a,b. Then, we identified marker metabolites of each cluster, Figure 2c.

### 3.3. Cluster Analysis Reflects Regional Differences and Pathological Spectra

In our cluster analysis, clusters 1, 2, 4, 5 and 7 revealed high intensities of N-acetyl-aspartate (NAA) and creatine (Cr), suggesting that this clusters predominantly contain normal appearing matter. Most of these voxels were also segmented as normal appearing matter, which confirmed our MRS data, Figure 2b–d. We computed the mean spectra of all voxels in those clusters, which reflected a normal configuration. The intensity of NAA was found to be the most variable parameter, Figure 2a. All other clusters revealed an altered pattern with various metabolic abnormalities, Appendix A. In cluster 8, the metabolic pattern was dominated by lactate and increased macromolecules, Figure 2a,c,d. This cluster contained voxels of the contrast-enhancement regions as well as necrotic tumor regions. The spectra showed high intensities at 1.3 ppm (lactate) and between 1–0.6 ppm (MM and lipids), which is characteristic of malignancy, Figure 2c. The next relative distant cluster 6 contained all ventricular segmented voxels and was marked by voxels low or absent signal intensity. Additionally, we found that voxels, which were segmented as FLAIR hyperintensity, were also enriched in this cluster, Figure 2b.

We assume that the voxels of FLAIR-hyperintense regions of cluster 3,9 and 10 significantly differ to the FLAIR-hyperintense voxels of cluster 6. We found that cluster 3 and 9 lack classical tumor-related metabolites, Appendix A, such as those present in cluster 8, suggesting that these FLAIR-hyperintensity within the voxels is most probably due to vasogenic edema rather than infiltrating high-grade tumor regions, Figure 3a,b.

### 3.4. Prediction of Tumor Regions

We trained a neural network with a subsequent linear discrimination model to predict the output variable: normal appearing matter, metastasis, glioma or inflammatory lesion, Figure 4a. We trained our model based on the segmented voxel (approx. 2/3 training and 1/3 validation, *n* = ~8000) and externally validated the findings within ~23,000 voxels from the whole dataset. We computed a prediction score which was fitted by a probability distribution model, centered and scaled. With a 97.3% probability of correctly classified voxels, the automated designation of pathological voxels is relatively robust, Figure 4b–d.

Prediction of the other subgroups revealed a lower accuracy for the whole model (83.43%). We used the spatial grid of MRS for tumor prediction and were able to reconstruct the tumor extension of the MRI, Figure 4e. In summary, normal and pathological MRS data can be distinguished with high accuracy.

### 3.5. Exploration of Metabolic Diversity in Pathological Lesions

In order to classify the heterogeneity of pathological voxels, we subsequently isolated the voxels that had been segmented as tumor or lesion and mapped the difference of metabolic profiles from other oncological and inflammatory brain diseases.

Several voxels could not be sharply separated between the different pathologies, except for non-Hodgkin lymphoma (NHL) and metastasis from non-small-cell lung carcinoma (NSCLC). Epidermoid tumors paired with ganglioglioma and pineal gland tumors. Voxels in the upper part of the dimensional reduction contained mostly benign lesions highly enriched NAAG, Figure 5a–c. Most interestingly, we found a highly significant enrichment of aspartic acid (Asp) in a lesion histologically defined as gliosis. Molecular subgroups of glioma showed a large overlap however high-grade glioma (IDH-wildtype glioma (glioblastoma)) stands out at the bottom part of the dimensional reduction map with increased intensities for lactate (Lac), lipids and macromolecules (MM). We found high intensities for myo-inositol (Ins) towards the areas assigned to NSCLC and NHL and glycerophosphocholine (GPC) significantly enriched for IDH-mutated with and without LOH (1p/19q-codeletion), Figure 5c.

### 3.6. Prediction Model for Molecular Tumor Subgroups

We applied an LDA model to train for separation between the molecular subgroups of glioma. By solely focusing on voxel from tumor regions, the model showed a high accuracy for prediction of wildtype or mutated glioma, with the lowest AUC in oligodendroglioma (76.77%). We noticed that metabolites strongly enriched in the normal-appearing matter are considerably dominant when distinguishing between glioma subgroups, leading to an interference in our model when all voxels are considered.

As shown in Figure 6a,b, the prediction score (tumor subgroups) is also relatively strong in the non-tumor voxels, Figure 6f, producing a large number of false positive predictions. To improve the overall predictive value, we built a two-layer model: Layer 1 predicted a segmentation mask of those voxels with a high probability of tumor content and in layer 2 the mask was used to focus on the subtype prediction of tumor voxels exclusively (layer 2), Figure 6e.

This model enables us to predict the molecular subgroup of patients by consideration of all pathological voxels, which resulted in a distribution of relative accuracy of all subgroups, Figure 6g. Using this model, we showed a 91.2% overall accuracy in predicting the molecular subtype of gliomas. Further, we provided the spatial distribution of each score, enabling the user to validate the predicted results and identify the tumor hotspot regions.

## 4. Discussion

In our prospective imaging study, we build an atlas of MRS profiles across tumor entities and molecular subgroups and designed a model for non-invasive prediction of glioma subgroups.

The need for improving the non-invasive diagnostic workup of brain tumors has been a crucial target in neuro-oncology over the last decades, especially since the rising importance of molecular characteristics for diagnostics and particularly for prognostic matters, with respect to expected treatment-response and survival. Modern technologies and artificial intelligence aid automated recognition and prediction of imaging in various medical applications, however the lack of feasibility hinders the embedding of modern algorithms in clinical routine. Particularly in neuroradiology, many studies on predictive models based on radiomic features have been conducted to facilitate the interpretation and implementation of imaging in daily routine, however, until now, the use of such techniques is not yet established [13]. We assume that metabolic information is more sensitive to detect molecular subgroups and so, we aimed to use the well-known MR spectroscopy and modern deep-learning algorithms for characterization of spectroscopic profiles. Recent studies have explored the ability of molecular subgroup prediction based on MRS in medulloblastomas [14,15], and a presurgical classification of pediatric cerebellar tumors based on metabolite ratios was successfully developed [7,16]. In adults, the same intent has been sought, however the integration of metabolite and genetic profiles of brain tumors and the translation of these techniques into the clinical practice has been challenging. In our study, we performed an unsupervised clustering of metabolic profiles with respect to their defined radiophenotype [17]. Clusters containing the majority of normal appearing matter were those with the highest intensities of NAA and creatine. NAA is a known metabolite of neurons and has been previously described in other studies as a marker for healthy neurons [18,19,20,21]. Regarding tumor diagnostics, it has been described to have elevated levels in pilocytic astrocytoma, which can be interpreted in this case as being closer to normal brain than to malignancy [6,16]. Creatine is a marker of intracellular metabolism and has usually higher levels in grey matter. Creatine was also found to be present in the normal appearing matter-related clusters along with NAA. Nevertheless, creatine is not a direct marker of healthy brain and usually the relevance to diagnostics of brain tumors is linked to a ratio along with choline, a more known metabolite associated with malignancy [16,18,22,23,24,25]. Recently, we investigated that creatine directly mediates resistance against hypoxic-stress in glioblastoma which causes a transcriptional subtype switch towards proneural gene expression [26]. The clusters with the highest levels of choline, lactate and macromolecules, characteristically present in cerebral hypoxia, inflammation, cellular membrane breakdown and necrosis [16,18,27,28,29], were the ones containing the voxels segmented to contrast-enhancement and necrotic tumor regions. We found that voxels, that were segmented as FLAIR-intensity, could be differentiated from vasogenic edema and tumor-infiltrating regions according to their metabolic profiles, with the first showing increased intensities of myo-inositol, which is a metabolite known as an “osmolyte” [30] and the last showing increased intensities of malignancy-related metabolites, such as macromolecules. These results are comparable to other studies conducted for preoperatively classification of brain tumors, in pediatric and adult population [6,7,15,16,28,31]. In our study, we analyzed and proved the accuracy with which MRS can discern between a brain lesion and normal brain tissue, which is in line with previous reports in the literature [3,7,24,28,29,31,32]. Moreover, our data supports that MRS is a valid tool that can separate between edema regions with a suppressed intensity (ventricle-like signature) or tumor-infiltrative regions. This is of high relevance in low-grade glioma tumors where contrast-enhancement is low and a clear separation between resectable tumor and non-resectable edematous brain tissue can be difficult based solely on MRI, which is particularly relevant from a surgical point of view.

## 5. Conclusions

In contrast to other published models which mainly used radiomic features, we have developed a classification algorithm combining a deep autoencoder and linear discriminant models that can predict the origin of a brain lesion (either metastatic or glial) based on metabolic profiles. In addition, within gliomas, our tool is able to determine the tumors type based on the metabolic profile with an acceptable accuracy rate allowing the prediction of prognosis-relevant parameters such as IDH-mutation and 1p19q-codeletion in a non-invasive manner. This is a first step into the integration of various aspects of the oncological treatment planning with the idea of combining the metabolic and molecular traits of lesions and improve personalized tumor treatment.

## Figures and Tables

**Figure 1 cancers-13-02417-f001:**
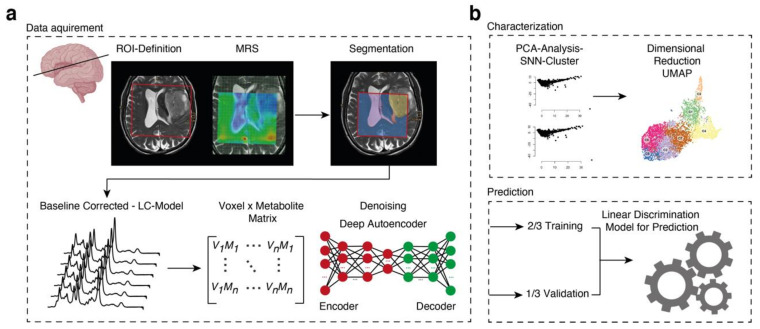
(**a**) Workflow for post-hoc data analysis. After segmentation and baseline correction, we performed a deep autoencoder for data denoising of fitted metabolites. (**b**) Characterization: the analysis was split into a first part in which data were deeply characterized including dimensional reduction and clustering, and a second part where a novel prediction model was established to allow precise prediction of the molecular subgroups through MRS.

**Figure 2 cancers-13-02417-f002:**
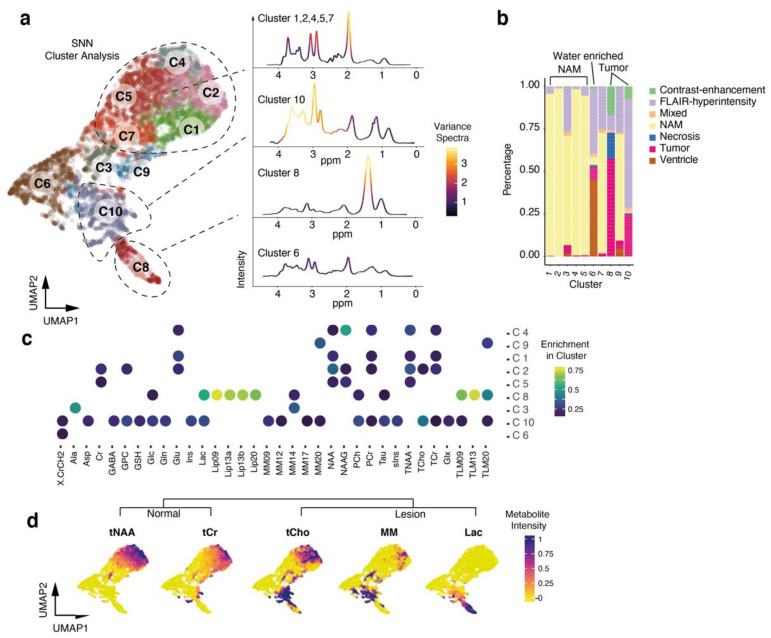
(**a**) 2D representation of a UMAP dimensional reduction. Colors illustrate the SNN clusters (1–10). (**b**) Bar plot indicates the distribution of segmented voxel within the cluster analysis. (**c**) Dot plot of significantly enriched metabolites of each cluster. Colors indicate the global intensity of each metabolite. (**d**) 2D representation of a UMAP dimensional reduction. Colors illustrate the metabolite intensity and marker metabolites of each cluster.

**Figure 3 cancers-13-02417-f003:**
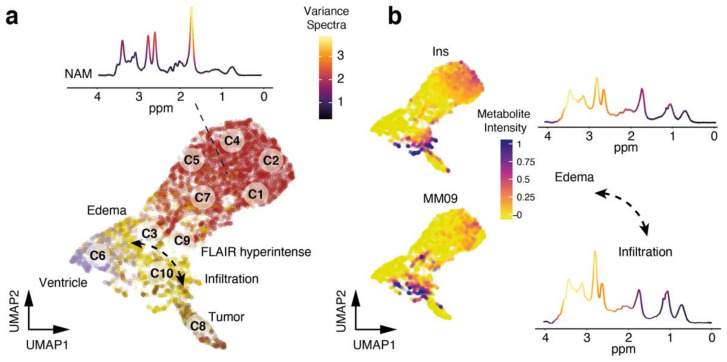
(**a**) 2D representation of a UMAP dimensional reduction. Colors illustrate the regions, red: normal-appearing matter (NAM), yellow: FLAIR-hyperintensity, brown: tumor, and purple: ventricle. FLAIR-hyperintensity regions are partly assigned to the ventricle regions (which contained a ow signal) and the tumor clusters, illustrated by an arrow. (**b**) On the left: 2D representation of a UMAP dimensional reduction. Colors illustrate marker metabolites myo-inositol (Ins), highly enriched in an edema region and a macro-molecule band (09), highly enriched within tumor-infiltrative regions compared to edema-related FLAR hyperintensity. On the right: Representative spectra of the FLAIR-hyperintensity edema region (top spectra) and FLAIR-hyperintensity tumor infiltration-related voxels (bottom spectra).

**Figure 4 cancers-13-02417-f004:**
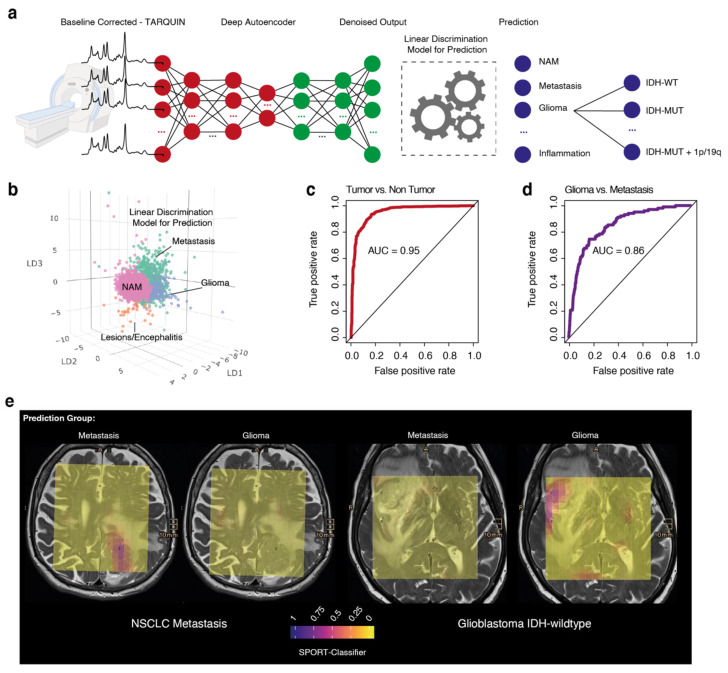
(**a**) Workflow of our prediction model. (**b**) 3D map of group prediction using an LDA. Colors indicate the subgroups: purple: normal-appearing matter, orange: pathological lesion (non-tumor pathology), blue: glioma and green: metastasis. (**c**,**d**) Representative presentations of the AUC from our prediction model. (**e**) Prediction maps of two patients, a metastasis (**left**) and an IDH-wildtype glioblastoma (**right**). Each voxel is colored based on its probability to contain voxels of the predicted entity. Yellow voxels represent low prediction scores, red-purple voxels represent higher prediction scores. The color scheme is plasma.

**Figure 5 cancers-13-02417-f005:**
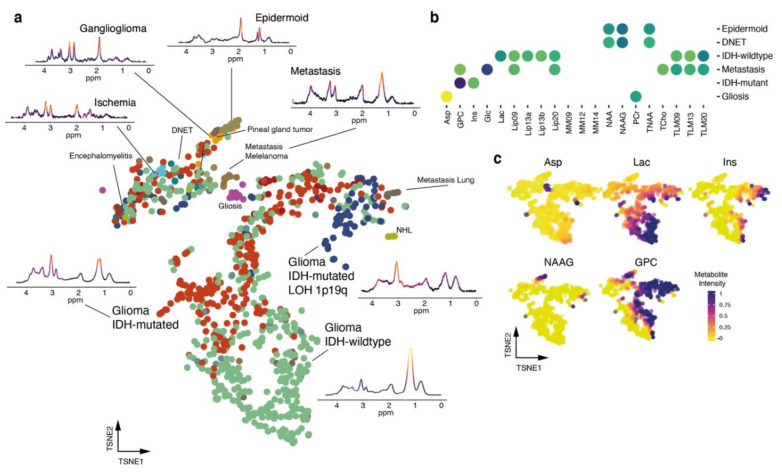
(**a**) 2D representation of a UMAP dimensional reduction of pathological confirmed lesions, representative spectra are shown. (**b**) Dot plot of significantly enriched metabolites for each representative brain disease type. (**c**) 2D representation of a UMAP dimensional reduction. Colors illustrate metabolite intensity.

**Figure 6 cancers-13-02417-f006:**
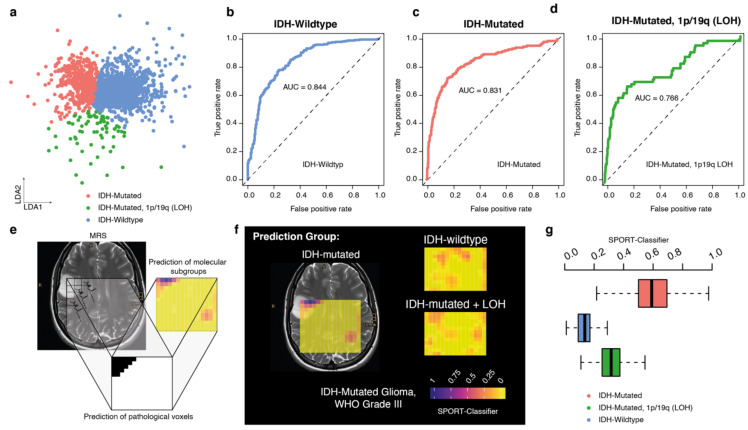
(**a**) 2D representation of an LDA map of gliomas. (**b**–**d**) AUC representations of the predictive value of each glioma subtype. (**e**,**f**) Example SPORT prediction model of glioma subtypes in one patient. Each voxel is colored based on its probability to contain voxels of the predicted entity. Yellow voxels represent low prediction scores, red-purple voxels represent higher prediction scores. The color scheme is plasma. (**g**) Representative SPORT classifier by summarizing all tumor voxels from level 1 prediction (layer 1) into a distribution of probabilities. The presented boxplot indicates the mean prediction score of each tumor voxel (from layer 1). Here, the patient’s classifier showed the highest score for the IDH-mutated subgroup.

## Data Availability

The data presented in this study are available from the corresponding author on reasonable request.

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
