# Peer review of "Mapping of Metabolic Heterogeneity of Glioma Using MR-Spectroscopy"

_cancers, 2021, doi:10.3390/cancers13102417_

Round 1
Reviewer 1 Report
In the present paper, the authors report on a prospective imaging trial in patients with brain tumors. They use MR imaging and MR spectroscopy data to describe and predict different brain tumor types. Their radiomics-based approach is innovative and driven by advanced computational methods that draw upon image analysis, dimensionality reduction and prediction. Their resulting model is very accurate and provides novel insights into the metabolic landscape of the different brain tumor types and across their topographic subregions.
The overall sample size seems adequate (although subgroups of rare tumor subtypes may be insufficiently powered) and the figures very professional.
I cannot provide critical feedback on the computational methods chosen since this is not my specialty, which may, however, be critical.
Author Response
Thank you very much for your comments, we really appreciate the time spent reviewing our manuscript.
Reviewer 2 Report
The title of the present manuscript suggest mapping of metabolic heterogeity of glioma using MR-spectroscopy. The authors conclude that in 91.2% of cases a molecular subtype could be predicted.
- The authors should describe the methods in such a way that a novice from neuro-oncology can follow what was done.
- The authors should discuss in more detail the metabolic heterogeneity
- were th 91.2 % of cases predicted on the basis of IDH1 wildtype/mutation?
- Why could the remaining 8.8% not be classified
- The biologic basis of lactate peaks and IDH1 mutations should be explained in more details
- How was 1p19q codeltion interpreted in terms of MRS peak changes?
- Did the authors attempt to use the "classical" histologic diagnostic groups like astrocytome, oligodendroglioma for correlation with MRS data instead of only using IDH1 mutated/wildtype glioma
Reviewer 3 Report
This report describes the combined use of MR spectroscopic imaging (termed CSI here) with a neural-network based classifier for discrimination of brain lesions that includes information on molecular subtypes. The report is of interest and while previous several reports have described similar tissue classification methods the inclusion of molecular subtyping adds novelty to this report. While the report is generally well written it nevertheless needs considerable improvement in the writing to correct awkward and frequently imprecise language. There are also some relatively minor grammar issues (e.g. incorrect use of commas). There is also a need for additional experimental details and clarification of the methods. There are also concerns on the spectroscopic data analysis.
Comments by line number:
L89 and methods. The description of the CSI method is incomplete. It needs the number of phase encodings. Also, the spatial resolution given needs to be described as the “nominal” resolution (I assume), i.e. as defined by the FOV and encodings.
L110 – How was non-contrast enhancing tumor identified? This is not listed in the methods section.
L 113 and figure. Clarify what is meant by “denoising”. In the MRS community this is understood to mean denoising of the spectrum, but here it seems to be referring to the data obtained after spectral fitting. However, the Figure 1a shows a stack of spectra, but then the matrix is of #metabolites in size, not the number of points in the spectrum. This same sequence is shown in Fig 4a, but it doesn’t seem correct that the spectra end up going into the discriminate model, as shown. These figures need clarifying.
L120. Need reference for UMAP.
L133, 135. I think “sum spectrum” would be better written as – the average spectrum …
Fig 2a. PPM scale is off. Also on other figures. It would be useful to show spectra from all clusters, add clusters 3 and 9.
All Figures. The use of multiple sections to each figure, each showing different information, is not recommended – it is confusing and ends up with each frame being too small. Figure section 1a certainly warrants being a separate figure. If necessary, these can be put in supplementary material.
L168 & Tables. See journal format requirements for decimals and thousands.
Fig 6f. Unclear what the images with IDH-wildtype and mutated labels are showing. Both show no significant classifications.
L268 – it is MYO-inositol
L 280. I would recommend in the closing or first paragraph of the discussion, making a clear statement as to what the major findings of this study are.
L311 Please check FLAIR SPACE. This may be correct, but I am more familiar with SPACE-FLAIR.
L316. Was CSI done before or after contrast administration? Was any spatial or spectral smoothing applied.
CSI Processing. This needs a list of what metabolites were included in the fitting and how the basis functions were derived. Later on this same page, a number of 34 metabolites is mentioned, which is grossly excessive for the number that can actually be fit and raises concerns of overfitting. If 34 were used this needs justifying. An essential requirement is that there also needs to be a description of what quality criteria were applied to reject bad spectra and/or fitting.
L328. A description is needed for what “shiny” is.
L331. Why was contrast enhancing tumor and necrosis grouped together? This list is different from that given earlier, please clarify these descriptions.
L334. Re: denoising - IF this is of the fitted metabolite results then please give a justification for why this is done. Is the gaussian distribution for one study, or over all studies? Why is a Gaussian distribution expected? If you have an abnormality I would expect a non-Gaussian distribution.
Supplemental material
I see no reason to include the code, particularly as it is made available in github.
I see no value in showing the “shiny” app.
Examples where language needs to be improved…
L104 State what is meant by “characterization”
L105 _ What is “Full proton” spectra. By 91 MRS I assume you mean studies, not voxels.
L136 – disbalances (maybe abnormalities)
L168 – …at spatial resolution…. – don’t follow
L185 – diversity -> difference
L187 – a frequent number – rephrase
L193 – distinct enrichment – clarify, enrichment of what?
L280 – other launched models – rephrase
L408 estimated -> evaluated …as …
Author Response
Response to Reviewer 3 Comments
- L89 and methods. The description of the CSI method is incomplete. It needs the number of phase encodings. Also, the spatial resolution given needs to be described as the “nominal” resolution (I assume), i.e. as defined by the FOV and encodings.
Response 1: Thank you for your correction, we have included the nominal distribution with the FOV and number of phase encoding steps in the “Methods” section. The description is also seen in the “Segmentation Tool in “Shiny” for MRS Data” of the Supplementary Methods.
- L110 – How was non-contrast enhancing tumor identified? This is not listed in the methods section.
Response 2: Non-contrast enhancing tumor was defined as hyperintensity on T2 or FLAIR. We didn’t define edema on MR imaging. This was added to the “Methods” section.
- L 113 and figure. Clarify what is meant by “denoising”. In the MRS community this is understood to mean denoising of the spectrum, but here it seems to be referring to the data obtained after spectral fitting. However, the Figure 1a shows a stack of spectra, but then the matrix is of #metabolites in size, not the number of points in the spectrum. This same sequence is shown in Fig 4a, but it doesn’t seem correct that the spectra end up going into the discriminate model, as shown. These figures need clarifying.
Response 3: Thank you for your comment. In our study, denoising is not related to the raw spectra but serves to reduce data noise within the already fitted metabolic data. After fitting metabolic intensities, we only get a partially blurred picture of the actual distribution of metabolites. This is due to possible technical reasons, batch effects within the cohort and other confounding factors. In addition, as noted in a later comment, there is the possibility of potential overfitting (despite exclusion of spectra with Q>1). Similar to the analysis of images, an autoencoder can be used to reduce potential blurring within metabolic data. The principle here is that we reduced and bundled the data step by step within a neural network. The learned network is then able to reconstruct the complete data based on the reduced information (within the bottleneck layers). We have been using this method for some time in the analysis of single cell sequencing and image analysis. However, it is also useful within metabolic data as we were able to show that the total variance of the first eigenvector (as an expression of the total variance of the data) can be increased using an autoencoder. A second advantage of this method is that the reconstructed data are normally distributed (this is due to the reduction of dropouts in the data set) which allows the application of linear prediction models.
- L120. Need reference for UMAP.
Response 4: We have included the respective reference for this dimensional reduction tool.
- L133, 135. I think “sum spectrum” would be better written as – the average spectrum.
Response 5: Thank you for your suggestion, we have corrected this.
- Fig 2a. PPM scale is off. Also on other figures. It would be useful to show spectra from all clusters, add clusters 3 and 9.
Response 6: Thank you for noticing, we have corrected this in all figures. We added the sample spectra of clusters 3 and 9 in the supplementary material.
- All Figures. The use of multiple sections to each figure, each showing different information, is not recommended – it is confusing and ends up with each frame being too small. Figure section 1a certainly warrants being a separate figure. If necessary, these can be put in supplementary material.
Response 7: We appreciate your opinion, however the use of complex figures is a common style in scientific papers which wish to accurately portray their results and is in accordance with this journal’s format and customs, as can be seen in several previously published studies.
- L168 & Tables. See journal format requirements for decimals and thousands.
Response 8: Thank you for noticing, we have corrected this along the entire manuscript.
- Fig 6f. Unclear what the images with IDH-wildtype and mutated labels are showing. Both show no significant classifications.
Response 9: We regret that this figure couldn’t be easily understood. What the figure shows is the use of our model to predict all glioma subgroups. The classifier is only positive for the actual tumor. An average of all voxels detected as tumors is then shown in the boxplot. We have clarified this in the figure.
- L268 – it is MYO-inositol.
Response: This has been corrected.
- L 280. I would recommend in the closing or first paragraph of the discussion, making a clear statement as to what the major findings of this study are.
Response: We added a summary statement at the start of our discussion.
- L311 Please check FLAIR SPACE. This may be correct, but I am more familiar with SPACE-FLAIR.
Response: This has been corrected.
- L316. Was CSI done before or after contrast administration? Was any spatial or spectral smoothing applied.
- Response: CSI was performed after contrast administration, hence the order of each imaging protocol in the “Imaging Acquisition and Data” of the “Methods” part. Neither spatial nor spectral smoothing was applied.
- CSI Processing. This needs a list of what metabolites were included in the fitting and how the basis functions were derived. Later on this same page, a number of 34 metabolites is mentioned, which is grossly excessive for the number that can actually be fit and raises concerns of overfitting. If 34 were used this needs justifying. An essential requirement is that there also needs to be a description of what quality criteria were applied to reject bad spectra and/or fitting.
Response 14: As commonly used, we determined the quality of fitting (Quality of the fit (Q)) by the standard deviation of the residual signal divided by the standard deviation of the spectral noise (SN). The SN is estimated by binning the residual signal into equally sized intervals in which the standard deviation (SD) of the signal was calculated and the bin with the smallest SD was then used as the noise estimator (1). We added this information into the “Methods” section.
(1) Wilson M, Reynolds G, Kauppinen RA, Arvanitis TN, Peet AC (2011) A Constrained Least-Squares Approach to the Automated Quantitation of In Vivo H-1 Magnetic Resonance Spectroscopy Data. Magnetic Resonance in Medicine 65: 1-12.
- L328. A description is needed for what “shiny” is.
Response 15: In “R”, “shiny” is a package that allows to build interactive web apps. This has been specified.
- L331. Why was contrast enhancing tumor and necrosis grouped together? This list is different from that given earlier, please clarify these descriptions.
Response 16: Thank you for noticing, we corrected the mistake.
- L334. Re: denoising - IF this is of the fitted metabolite results then please give a justification for why this is done. Is the gaussian distribution for one study, or over all studies? Why is a Gaussian distribution expected? If you have an abnormality I would expect a non-Gaussian distribution.
Response 17: As explained above, it is not a matter of forcing all metabolites within all subgroups or clusters into a normal distribution. Rather, it is natural that in large data sets the metabolites be normally distributed in their entirety (within all voxels). This does not mean that this also applies to voxels in pathological conditions.
- Supplemental material. I see no reason to include the code, particularly as it is made available in github. I see no value in showing the “shiny” app.
Response 18: We really appreciate your opinion; however, we have decided to still include the code and the shiny app in order to support the reproducibility of our methods.
- Examples where language needs to be improved…
L104 State what is meant by “characterization”
L105 _ What is “Full proton” spectra. By 91 MRS I assume you mean studies, not voxels.
L136 – disbalances (maybe abnormalities)
L168 – …at spatial resolution…. – don’t follow
L185 – diversity -> difference
L187 – a frequent number – rephrase
L193 – distinct enrichment – clarify, enrichment of what?
L280 – other launched models – rephrase
L408 estimated -> evaluated …as …
- Response: We have corrected these mistakes.

Reviewer 4 Report
The manuscript reports very interesting 2D CSI MRS results. The way they analyze the MRS data are novel. A major concern is small sample size to make any meaningful conclusion. Due to large voxel size, data on metastasis may not be reliable, not to mention small sample size. What is the percent of metastatic tumors in the voxels they claim from metastasis? In figure 6, they only sub-categorized into IDHwt, IDHmut, and IDHmut/1p/19q codel, but grades also need to be included, as tumor cell density is very different for each grade.
Reviewer 5 Report
The paper proposed an AI approach to analyzing MR spectroscopy data for prediction and classification of spectroscopic profiles. This is a novel application of deep learning to detecting and analyzing brain tumors. The paper describes the neural network components clearly, and motivates the problem. However, I found the goal a bit hard to follow. It would help to describe the main idea in the introduction, and better tie in the components of the system into Figure 1. Please also describe the used hyper-parameters (learning rate, optimizer) neural network architecture, number of examples in the train/testing data splits in a dedicated section.
Why do the authors perform LDA on the denoised features instead of standard classification (Multi-class, if necessary) using a neural network? Is it possible to have an end-to-end system? Also, why is shared nearest neighbor (SNN) used, rather than PCA or tSNE for clustering?
The idea is clearly thought through, but I would recommend streamlining it.
Text comments:
- Add link to codes in a footnote
- Line 137: “gaussian ” -> Gaussian
- “Ee” (line 208) what is this?
Round 2
Reviewer 4 Report
The revised version is much better.
Author Response
Dear Reviewer, we are very thankful for your positive feedback regarding the revised version of our manuscript.